# Prevalence of Fatigue and Unrecognized Depression in Patients with Inflammatory Bowel Disease in Remission under Immunosuppressants and Biologicals

**DOI:** 10.3390/jcm10184107

**Published:** 2021-09-11

**Authors:** Marie Truyens, Elodie De Ruyck, Gerard Bryan Gonzales, Simon Bos, Debby Laukens, Martine De Vos

**Affiliations:** 1IBD Research Unit, Department of Internal Medicine and Pediatrics, Ghent University, 9000 Ghent, Belgium; marie.truyens@ugent.be (M.T.); simon.bos@ugent.be (S.B.); martine.devos@uzgent.be (M.D.V.); 2VIB Center for Inflammation Research, 9052 Ghent, Belgium; 3Department of Gastroenterology, University Hospital Ghent, 9000 Ghent, Belgium; 4Department of Gastroenterology, AZ Nikolaas, 9100 Sint-Niklaas, Belgium; elodie.deruyck@aznikolaas.be; 5Nutrition, Metabolism and Genomics Group, Division of Human Nutrition and Health, Wageningen University, 6700 WE Wageningen, The Netherlands; bryan.gonzales@wur.nl

**Keywords:** Crohn’s disease, ulcerative colitis, behavior, depression, *NOD2*, Lipocalin-2

## Abstract

Background: Although highly prevalent among inflammatory bowel disease (IBD) patients, fatigue remains an unmet clinical need. The aim was to describe the prevalence of fatigue in an IBD population in remission and identify factors associated with fatigue. Methods: IBD patients in clinical and biochemical remission under treatment with immunomodulators or biologicals were included. Fatigue, physical tiredness and depression were assessed using the fatigue Visual Analogue Scale (fVAS), the Shortened Fatigue Questionnaire (SFQ) and the Quick Inventory of Depressive Symptomatology-Self Report (QIDS-SR), respectively. Relevant clinical and biochemical parameters were included in regression analyses to identify factors associated with physical fatigue. Results: In total, 157 IBD patients were included. Up to 45.9% of patients reported fatigue, physical tiredness was observed in 51% and depression in 10.8%. The majority of patients with subclinical depression were fatigued. Female sex (OR = 4.17 [1.55–6.78], *p* = 0.002) was independently associated with physical fatigue. Transferrin saturation (OR = −0.11 [−0.22–−0.007], *p* = 0.037) and treatment with adalimumab (compared to infliximab, OR = −3.65 [−7.21–−0.08], *p* = 0.045) entailed a lower risk of fatigue. Conclusion: Fatigue is observed in about half of IBD patients in remission and can be a symptom of underlying undetected depression. Sex, transferrin saturation and medication were identified as independent risk factors.

## 1. Introduction

Inflammatory bowel diseases (IBDs) are chronic inflammatory conditions of the gastrointestinal tract. The major disease entities are Crohn’s disease (CD) and ulcerative colitis (UC), characterized by episodes of remission and flare-ups [1,2,3]. The quality of life (QoL) in IBD patients is determined by more than gastrointestinal symptoms alone; disabling fatigue is often reported by patients as one of the most burdensome symptoms, especially during disease remission, when gastrointestinal symptoms are less dominant over daily life [4,5,6,7,8]. The estimated prevalence of fatigue is 72% during active intestinal inflammation and 47% during remission [9]. Another frequently neglected health problem in IBD is depression. If left untreated, mental disorders in IBD patients are linked with more severe symptoms, more flares, higher hospitalization rates and lower treatment compliance [10].

Currently, the multifactorial pathophysiology of fatigue is incompletely understood. Factors such as inflammation, nutritional deficiencies, metabolic alterations, psychological comorbidities, lifestyle and gut microbiota composition are thought to play a role [6]. In addition, recent evidence indicates that Nod-like receptors, including *NOD2*, which is one of the most notorious genes associated with IBD [11,12], are regulators of gut-to-brain signaling. Genetic deletion of *Nod1* and *Nod2* in mice resulted in behavioral abnormalities, with underlying stress-induced hyperactivation of the hypothalamic–pituitary–adrenal axis and impairments of the serotonergic system in the brain [13]. Lipocalin-2 (LCN2) is a glycoprotein that mediates multiple inflammatory processes and is elevated in serum of patients with active IBD, but decreases during periods of disease remission [14,15,16]. Recent articles have shown a potential role for LCN2 in the induction of neuroinflammation associated with obesity and nonalcoholic steatohepatitis [17,18]. Moreover, genetic deletion of LCN2 resulted in decreased fatigue behavior in mice with lipopolysaccharide-induced neuroinflammation [19]. The primary objective of this study was to determine the prevalence of fatigue and its correlation with depression in an IBD population in remission. The secondary objective was to report possible clinical and biochemical factors associated with fatigue, including *NOD2* variants and serum LCN2 levels.

## 2. Materials and Methods

### 2.1. Study Population

The study population consisted of adult IBD patients, either CD or UC, followed at a single tertiary referral hospital. In order to assess a population with comparable disease severity, only patients under treatment with immunosuppressants (IS) (azathioprine, methotrexate or mercaptopurine) and/or biologicals (infliximab (IFX), adalimumab (ADA) or vedolizumab (VDZ)) for at least 4 months were recruited. Stable clinical and biochemical remission for the last 6 months was required. Clinical remission was defined as a Harvey–Bradshaw Index (HBI) ≤ 3 for CD patients and a clinical Mayo score ≤ 1 for UC patients [20,21]. Biochemical remission was defined as C-reactive protein (CRP) < 10 mg/L. Patients who had an active flare and/or steroid need in the last 6 months and patients with clinical depression who were under treatment with antidepressants or with other significant psychiatric comorbidities were excluded. Other exclusion criteria were malignancy (active or in the personal history) and anemia (defined as a hemoglobin < 11.8 g/dL).

### 2.2. Assessment of Fatigue and Depression

The following questionnaires were used: fatigue Visual Analogue Scale (fVAS), Shortened Fatigue Questionnaire (SFQ) and the Quick Inventory of Depressive Symptomatology-Self Report (QIDS-SR). The fVAS was chosen because of its good performance and correlation with longer scales, with good applicability in routine clinical care [22]. Following international standards, the cut-off for fatigue was set at fVAS ≥ 5 [23]. The SFQ is an easy-to-perform questionnaire that assesses the intensity of the physical fatigue of the previous 2 weeks. The score varies from 4–28, and the cut-off for fatigue was set at ≥18 [24]. The SFQ was selected since it has excellent reliability (Cronbach-α: 0.88) and is easy implementable; however, it was not validated in IBD [25]. To assess depression, the QIDS-SR was used, in which the total score varies from 0 to 27; the cut-off for depression was set at ≥11 [26,27].

### 2.3. Laboratory Analyses and NOD2 Genotyping

Blood was collected for hemoglobin, iron and CRP assessment at inclusion. LCN2 levels were assessed using a Magnetic Luminex^®^ Performance Assay for human LCN2 (R&D Systems, Minneapolis, MN, USA). Genomic DNA isolation was performed from whole blood of patients who provided consent (*n* = 151). *NOD2* variants rs2066844, rs2066845 and rs2066847 were genotyped using Taqman assays (Thermo Fisher Scientific, Life technologies Corporation, Waltham, MA, USA), following the manufacturer’s instructions.

### 2.4. Statistical Analysis

SPSS version 27 (IBM Corp. Released 2020. IBM SPSS Statistics for Windows, Version 27.0, Armonk, NY, USA) and GraphPad Prism^®^ (GraphPad Software Inc., La Jolla, CA, USA) software packages were used for statistical analysis and graphical representations of data, respectively. Data with normal distribution and non-normal distribution are presented with the mean and standard deviation or the median and interquartile range (IQR), respectively. The Spearman’s rho correlation coefficient (r_s_) was determined to identify correlations between the different scores, whereas univariate and multivariate linear regression analyses were performed to identify risk factors associated with fatigue. Since there was no difference between IFX and VDZ, whether combined with an immunosuppressant or not, these were grouped together to augment the power. *p*-values for univariate analyses were corrected using false discovery rate (FDR) adjustment [28]. For the multivariate regression analysis, 6 clinical hypothesis-driven linear models were tested in order to select the model with the best parameters. Multiple models were compared based on the Akaike Information Criterion (AIC), *p*-value and adjusted R^2^ (Appendix A). Multicollinearity was excluded (all tolerance values > 0.4 and variance inflation factors < 2.5). Statistical significance was defined as a *p*-value < 0.05.

## 3. Results

### 3.1. Characteristics of the Study Population

A total of 318 patients were screened for participation; patients’ records were reviewed to verify that patients met all the inclusion criteria. At the time of inclusion, an HBI or clinical Mayo score was filled out to verify clinical remission. Based on the inclusion and exclusion criteria, 157 patients (78.3% CD and 21.7% UC) were included. Patient characteristics are summarized in Table 1. The percentage of women and men participating was equal. The majority of patients (83.4%) were treated with biologicals, mainly IFX. The median disease duration in this cohort was relatively long: 11 years (IQR 6–18.5). All patients were in clinical and biochemical remission; the majority of patients had an HBI or clinical Mayo score of 0 and a low median CRP and LCN2 of 1.4 mg/L (IQR 0.3–3.4) and 23.2 ng/mL (IQR 14.5–32.2), respectively. Signs of iron deficiency were present in around 20% of patients: 18.5% had a ferritin of <30 µg/L and 24.2% had a transferrin saturation (TSAT) lower than 20% without overt anemia in this cohort. The questionnaires were taken throughout the year: 21%, 35%, 21% and 22.9% were completed in the spring, summer, fall and winter, respectively.

### 3.2. Correlations between the Patient-Reported Outcomes Related to Fatigue and Depression

Overall, significant correlations were observed between the patient-reported outcomes related to fatigue and depression (fVAS, SFQ and QIDS-SR). The strongest correlation was observed between the fVAS and SFQ (r_s_ = 0.81, *p* < 0.001), while correlations were 0.549 between fVAS and QIDS-SR (*p* < 0.001) and 0.593 between SFQ and QIDS-SR (*p* < 0.001, Figure 1). In patients with fatigue (SFQ ≥ 18), 20% showed signs of depression (QIDS-SR ≥ 11), compared to 1.3% in patients without fatigue (*p* < 0.001). All but one patient with a QIDS-SR ≥ 11 reported significant fatigue (16/17, 94.1%).

### 3.3. Fatigue and Depression Scores and Factors Associated with Fatigue

Fatigue (fVAS ≥ 5) was reported by 45.9% of IBD patients with a comparable mean fVAS score in UC and CD (Table 2). Physical tiredness (SFQ ≥ 18) was reported in 51% of the patients (58.8% in UC and 48.8% in CD). Clinical unsuspected moderate to severe depression (QIDS-SR ≥ 11) was observed in 10.8% of patients.

Univariate linear regression analyses were performed for physical fatigue (SFQ ≥ 18, Table 3) with different demographic and biochemical variables. After FDR correction, only female sex was significantly associated with fatigue (OR = 3.88 [1.84–5.92], *p* < 0.001). This trend was seen in all scores, and fVAS and SFQ were significantly higher in women compared to men (Figure 2). The other assessed variables were not associated with fatigue in the univariate regression.

For the multivariate analysis, multiple models were compared, after which the best model was selected to assess factors contributing to physical fatigue (Appendix A). The final model had an adjusted R^2^ of 0.141, AIC of 425.83 and *p* = 0.003. In this multivariate analysis, female sex remained a significant risk factor for physical fatigue (OR = 4.17 [1.55–6.78], *p* = 0.002, Table 3). Higher TSAT was also significantly associated with lower fatigue (OR = −0.11 [−0.22–−0.007], *p* = 0.037). Considering the type of medication, IFX was associated with higher SFQ. Patients treated with ADA reported significantly lower fatigue scores compared to IFX (OR = −3.65 [−7.21–−0.08], *p* = 0.045). Other demographic and clinical variables, i.e., duration of disease, *NOD2* variants, previous resection and LCN2 concentrations were not significantly associated with physical fatigue (Table 3). Parameters such as BMI, smoking behavior, age, duration of current treatment, ferritin and season were tested in the different models, without significant association with fatigue, and were not withheld for the final model (Appendix A).

## 4. Discussion

Fatigue remains an unmet clinical need among IBD patients due to the incomplete understanding of its pathogenesis and the lack of targeted therapeutic strategies. Fatigue is highly prevalent during active disease but often persists during disease remission, with a more severe impact on the QoL [7,8,23,29].

The current study reported abnormal physical fatigue (SFQ ≥ 18) in 51% of IBD patients in well-defined clinical and biochemical remission. These results are comparable with studies from Villoria et al., where 54% of IBD patients in remission were fatigued (based on the Functional Assessment of Chronic Illness Therapy-Fatigue (FACIT-F)), and Varbobitis et al., with a fatigue rate of 46% (IBD-Fatigue scale) [30,31]. In the studies by Williet et al. (FACIT-F), Bager et al. (Multidimensional Fatigue Inventory (MFI)) and Graff et al. (MFI) the prevalence of fatigue was slightly lower: around 26–30% of IBD patients in remission [29,32,33]. Globally, these rates are significantly higher compared with the general population, where fatigue is reported in 5–10% [23,33]. A good correlation was observed between the SFQ and fVAS, as seen in other studies comparing the fVAS to longer questionnaires, indicating that the fVAS can be implemented as a quick screening tool to identify fatigued patients during IBD consultations [22,23].

In line with the available literature, female sex was one of the main risk factors for fatigue, but a good explanation for this observation has not yet been found [30,31,32,34,35]. The effect of TNF-inhibitors on fatigue is still undetermined. Evidence for improvement of fatigue in patients with active disease treated with IFX (or other biologicals) was reported [35,36]. However, other studies reported a higher prevalence of fatigue in patients treated with TNF-inhibitors, considering that fatigue, as a side-effect, occurs in 9% of patients on IFX [30,37,38]. Interestingly, the present study showed a significantly lower degree of fatigue in patients treated with ADA when compared to IFX treatment.

Since fatigue can be a symptom of anemia, low hemoglobin was an exclusion criterion for this study. To assess the effect of iron on fatigue, the relationship between iron parameters and physical fatigue was assessed: higher TSAT levels were significantly associated with lower fatigue. To date, no straightforward association between iron deficiency and fatigue is reported in non-anemic IBD patients. However, it is generally recommended to screen for iron deficiency and anemia, and treat if necessary, which is confirmed by these results [4,33,39]. LCN2 concentrations were very low in this cohort, indicating a deep state of disease remission [14,16]. Although reported as a possible inductor of neuroinflammation, no correlation between LCN2 and fatigue (nor depression) was found in this study.

A seasonal impact on general functioning is described in the general population as well as in IBD, with symptoms such as loss of energy or worsening of mood during winter [40]. In the current study population, there was no significant seasonal impact on fatigue or depression. Additionally, other factors such as *NOD2* variants, concurrent EIMs, disease duration and previous resection were not associated with fatigue in this study.

In addition to fatigue, depression is also known to decrease the QoL in IBD patients. Unfortunately, this psychological comorbidity is often under-researched and unrecognized in IBD [41], which in turn leads to suboptimal treatment and potential worse IBD-related outcomes [42,43]. The prevalence of depression in patients with IBD is reported to be almost double the prevalence in the general population [10,43]. The link between IBD and depression is thought to be bidirectional: the stressful experience of living with IBD might trigger psychiatric comorbidities or depression itself might intensify IBD disease activity [43,44]. Alternatively, dysregulated immunoregulatory circuits are implicated in the pathogenesis of both depression and IBD [44]. In the present study, clinically unrecognized depression was reported in 10.8%. This percentage is lower than the reported prevalence of depression in IBD patients (in remission) in the study by Zhang et al. (22%) because patients already diagnosed with and treated for depression were excluded from the present study [45]. As in fatigue, female sex was an important risk factor. Fatigue is a possible symptom of depression and an overlap between depression, fatigue and anxiety is described in literature [36,46]. Further analysis of the study population showed that 20% of patients reporting significant fatigue (SFQ ≥ 18) had concurrent moderate to severe depressive complaints, which is comparable to the results of Cohen et al. but lower than the prevalence of depressive symptoms found in the study by Borren et al. [35,47]. Since the vast majority of patients with signs of depression (QIDS-SR ≥ 11) were fatigued, the presence of fatigue should always trigger physicians to exclude underlying depression.

The current study had the following limitations and strengths: biochemical remission was determined as a CRP < 10 mg/L, which might be less sensitive than fecal calprotectin; the study was monocentric and rather small but included a well-defined IBD population; and the prevalence of depression was too low to permit thorough analyses.

## 5. Conclusions

In conclusion, in IBD patients in clinical remission under immunosuppressants and/or biologicals, fatigue is highly prevalent and can be a symptom of unsuspected depression. The main risk factors for fatigue are female sex, low TSAT and IFX treatment (compared to ADA).

## Figures and Tables

**Figure 1 jcm-10-04107-f001:**
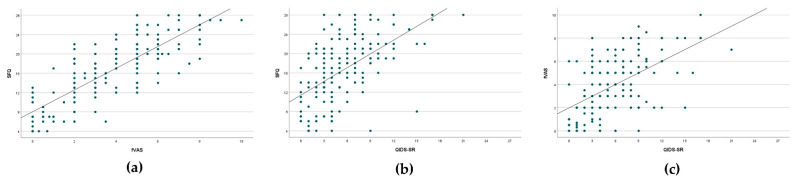
Correlation between the different scores assessing fatigue and depression. (**a**): Correlation between the Shortened Fatigue Questionnaire (SFQ) and the fatigue Visual Analogue Scale (fVAS), Spearman’s rho correlation coefficient r_s_ = 0.810, *p* < 0.001; (**b**): Correlation between the Quick Inventory of Depressive Symptomatology-Self Report (QIDS-SR) and SFQ, r_s_ = 0.593, *p* < 0.001; (**c**): Correlation between the fVAS and QIDS-SR, r_s_ = 0.549, *p* < 0.001.

**Figure 2 jcm-10-04107-f002:**
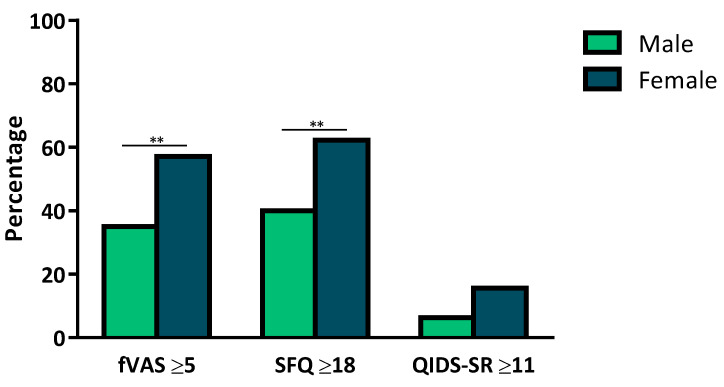
Graphic presentation of the fatigue and depression scales by sex. Differences were calculated using the Pearson χ2 test. Female versus male: fVAS ≥ 5: 57.1% vs. 35% (** *p* = 0.005), SFQ ≥ 18: 62.3% vs. 40% (** *p* = 0.005), QIDS-SR ≥ 11: 15.6% vs. 6.3% (*p* = 0.06). fVAS: fatigue Visual Analogue Scale; SFQ: Shortened Fatigue Questionnaire; QIDS-SR: Quick Inventory of Depressive Symptomatology-Self Report.

**Table 1 jcm-10-04107-t001:** Characteristics of the study population.

Variables	UC *n* = 34 (21.7%)	CD *n* = 123 (78.3%)	Total *n* = 157
Age (years), median (IQR)	47 (38–60)	39 (29–50)	41 (30–53)
Sex, *n* (%)			
Female	21 (61.8)	56 (45.5)	77 (49.0)
Male	13 (38.2)	67 (54.5)	80 (51.0)
BMI (kg/m^2^), mean (SD)	25.1 (4.2)	24.6 (3.6)	24.7 (3.7)
Smoking, *n* (%)			
Never	16 (69.6)	70 (65.4)	86 (66.2)
Past smoker	6 (26.1)	16 (15.0)	22 (16.9)
Current smoker	1 (4.3)	21 (19.6)	22 (16.9)
Extra-intestinal manifestation (yes), *n* (%)	10 (30.3)	21 (17.1)	31 (19.9)
Arthralgia (axial/peripheral)	6 (18.2)	7 (5.7)	13 (8.3)
Arthropathy (axial/peripheral)	4 (12.1)	13 (10.6)	17 (10.9)
Uveitis	0	1 (0.8)	1 (0.6)
Disease duration (years), median (IQR)	9 (4–18)	11 (6–19)	11 (6–19)
Montreal score, *n* (%)	UC	CD	
	E1: 2 (5.9)	L1: 26 (21.1)	-
	E2: 15 (44.1)	L2: 16 (13)	-
	E3: 17 (50)	L3: 69 (56.1)	-
	-	L3 + L4: 12 (9.7)	-
History of resection (yes), *n* (%)	0	32 (26.0)	32 (20.4)
Hemoglobin (g/dL), mean (SD)	14.1 (1.3)	14.1 (1.3)	14.1 (1.3)
TSAT (%), mean (SD)	35.2 (14.2)	28.9 (12.0)	30.3 (12.7)
Ferritin (µg/L), median (IQR)	67 (26.5–151.5)	66 (38.5–124.5)	66.5 (36.8–128.0)
CRP (mg/L), median (IQR)	0.9 (0.3–1.3)	1.8 (0.7–3.6)	1.4 (0.3–3.4)
Lipocalin-2 (ng/mL), median (IQR)	18.3 (13.1–29.6)	24.4 (14.5–33.3)	23.2 (14.5–32.2)
*NOD2* variation (yes), *n* (%)	6 (19.4)	42 (35.0)	48 (31.8)
Previous lines of biologicals, *n* (%) (TNF inhibitors and/or vedolizumab)			
No previous biologicals	23 (69.7)	88 (71.5)	111 (71.2)
One	2 (6.1)	26 (21.1)	28 (17.9)
Two	7 (21.2)	8 (6.5)	15 (9.6)
Three	1 (3.0)	1 (0.8)	2 (1.3)
Current treatment, *n* (%)			
Immunosuppressant (IS)	4 (11.8)	22 (17.9)	26 (16.6)
Adalimumab	5 (14.7)	20 (16.3)	25 (15.9)
Infliximab	13 (38.2)	52 (42.3)	65 (41.4)
Infliximab + IS	4 (11.8)	9 (7.3)	13 (8.3)
Vedolizumab	6 (17.6)	16 (13.0)	22 (14.0)
Vedolizumab + IS	2 (5.9)	4 (3.3)	6 (3.8)
Duration current treatment (months), median (IQR)	35 (11–69)	32 (12–85)	33 (12–76)
Disease activity by score, *n* (%)	Mayo for UC	HBI for CD	
0	33 (97.1)	80 (66.1)	-
1	1 (2.9)	25 (20.7)	-
2		10 (8.3)	-
3		6 (5)	-

Body mass index (BMI): missing data *n* = 28; Smoking: missing data *n* = 27; Extra-intestinal manifestation (EIM): missing data *n* = 1; Transferrin saturation (TSAT): missing data *n* = 8; Ferritin: missing data *n* = 11; *NOD2* variant: the presence of at least 1 rs2066844, rs2066845 or rs2066847 minor allele; Duration current treatment: missing data *n* = 1; Clinical Mayo score in UC patients, cut-off for remission and inclusion was ≤3. No missing data; Harvey–Bradshaw index (HBI) in CD patients, cut-off for remission and inclusion was ≤3. Missing data: *n* = 2.

**Table 2 jcm-10-04107-t002:** Fatigue and depression scores.

	UC *n* = 34	CD *n* = 123	Total *n* = 157	*p*-Value
fVAS, mean (SD)	3.9 (2.8)	3.9 (2.3)	3.9 (2.4)	
fVAS < 5, *n* (%) fVAS ≥ 5, *n* (%)	17 (50.0) 17 (50.0)	68 (55.3) 55 (44.7)	85 (54.1) 72 (45.9)	0.584
SFQ, mean (SD)	17.3 (7.6)	16.7 (6.5)	16.8 (6.7)	
SFQ < 18, *n* (%) SFQ ≥ 18, *n* (%)	14 (41.2) 20 (58.8)	63 (51.2) 60 (48.8)	77 (49.0) 80 (51.0)	0.300
QIDS-SR, mean (SD)	6 (4.4)	5.6 (4.0)	5.7 (4.1)	
QIDS-SR < 11, *n* (%) QIDS-SR ≥ 11, *n* (%)	30 (88.2) 4 (11.8)	110 (89.4) 13 (10.6)	140 (89.2) 17 (10.8)	0.764

Differences between UC and CD were calculated using the Pearson χ2 test or Fisher’s Exact test.

**Table 3 jcm-10-04107-t003:** Linear regression analyses of the SFQ score in the total population.

	Univariate OR [95% CI]	Adjusted *p*-Value	Multivariate OR [95% CI]	*p*-Value
Sex (female)	3.88 [1.84–5.92]	<0.001	4.17 [1.55–6.78]	0.002
Age	0.05 [−0.03–0.13]	0.269		
IBD type (CD)	−0.65 [−3.23–1.94]	0.734		
Duration of disease (≥20 years)	3.01 [0.47–5.56]	0.130	2.70 [−0.83–6.23]	0.132
Medication				
Infliximab	2.30 [0.20–4.40]	0.139	Reference	
IS in monotherapy	−0.10 [−2.97–2.77]	0.944	−0.27 [−3.74–3.21]	0.880
Adalimumab	−2.25 [−5.14–0.64]	0.269	−3.65 [−7.21–−0.08]	0.045
Vedolizumab	−1.78 [−4.54–0.99]	0.269	−1.05 [−4.50–2.40]	0.547
*NOD2* variation (yes)	0.32 [−2.03–2.67]	0.853	1.75 [−0.65–4.16]	0.774
History of resection (yes)	1.88 [−0.75–4.51]	0.269	2.67 [−0.49–5.82]	0.101
TSAT	−0.07 [−0.16–0.01]	0.269	−0.11 [−0.22–−0.01]	0.037
CRP	0.32 [−0.16–0.80]	0.269		
LCN2	−0.04 [−0.10–0.02]	0.269	−0.04 [−0.11–0.04]	0.350

OR = odds ratio. Univariate regression: False discovery rates (FDR) adjusted *p*-values. Multivariate regression model: adjusted R^2^ = 0.141, AIC = 425.83, *p* = 0.003.

## Data Availability

The data presented in this study are available on request from the corresponding author. The data are not publicly available due to ethical restrictions.

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
