# Peer review of "Prevalence of Fatigue and Unrecognized Depression in Patients with Inflammatory Bowel Disease in Remission under Immunosuppressants and Biologicals"

_jcm, 2021, doi:10.3390/jcm10184107_

Round 1
Reviewer 1 Report
IBD patients commonly experience fatigue. In this study, Truyens et al evaluated fatigue, physical tiredness and depression in IBD patients and found an association with fatigue and sex, transferrin saturation and medication (particularly Adalimumab). Interestingly, fatigue was not associated with iron parameters (anemia).This work fills a gap in overall knowledge and has the potential the impact the well-fare of IBD patients. Overall the manuscript is comprehensive and logical and the figures are excellent.
Author Response
Point 1: IBD patients commonly experience fatigue. In this study, Truyens et al evaluated fatigue, physical tiredness and depression in IBD patients and found an association with fatigue and sex, transferrin saturation and medication (particularly Adalimumab). Interestingly, fatigue was not associated with iron parameters (anemia).This work fills a gap in overall knowledge and has the potential the impact the well-fare of IBD patients. Overall the manuscript is comprehensive and logical and the figures are excellent.
Response 1: We would like to thank the reviewer for the positive feedback on our manuscript.
Reviewer 2 Report
- Great work done on an important but usually overlooked area in IBD field.
- Discussion: Would be more informative to compare the current study with more articles in terms of subclinical or unsuspected depression and the factors impacting that.
- Study Population: Considering current guidelines, endoscopic mucosal healing could be another factor to be considered as inclusion criteria for remission and Fecal Calprotectin would be a better surrogate for biochemical remission in the study population than CRP (which is mentioned in the study limitations). However, the current criteria would be good enough to include patients in remission into the study.
Author Response
Point 1: Discussion: Would be more informative to compare the current study with more articles in terms of subclinical or unsuspected depression and the factors impacting that.
Response 1: As suggested by Reviewer 2 we added more literature to the manuscript, supporting the importance of assessing depressive symptoms in IBD patients and the potential consequences of unrecognized depression in this population.
Point 2: Study Population: Considering current guidelines, endoscopic mucosal healing could be another factor to be considered as inclusion criteria for remission and Fecal Calprotectin would be a better surrogate for biochemical remission in the study population than CRP (which is mentioned in the study limitations). However, the current criteria would be good enough to include patients in remission into the study.
Response 2: We fully agree with the reviewer that it would have been interesting to include fecal calprotectin values and endoscopic data to define disease remission. However, the current study was aimed at assessing the fatigue levels in an unbiased, real-life IBD cohort followed at our tertiary unit. Since in the majority of patients no semi-recent (within 6 months before inclusion) calprotectin and endoscopic data were available, we could not include these data. However, we aimed at including a highly homogenous population of patients with clinical and biochemical remission based on strict clinical remission criteria, which was confirmed by biochemical parameters CRP and LCN2 concentrations in blood.
Reviewer 3 Report
Thank you for giving me the opportunity to read the manuscript entitled "Prevalence of fatigue and subclinical depression in patients with inflammatory bowel disease in remission: an observational monocentric study".
The work is interesting and adds some information on the issue of fatigue in the IBD population. I have some comments on the manuscript that are reported here below.
I don't understand why the authors selected only patients treated with immunosuppressants or biologicals; they do not represent the whole IBD population. These criteria should be changed or explained and the paper title changed accordingly. At first reading, it seems that authors used convenience sampling, like if the authors were able to recruit and follow-up only patients with these characteristics.
Did the authors perform CRP or HS-CRP? In my opinion, the latter should be better for IBD patients. Please add an explanation.
Intestinal inflammation biomarkers should be added to screen patients in order to confirm disease remission (e.g. Faecal Calprotectin or Lactoferrin). In my opinion, symptoms and blood tests are not enough. Talking about endoscopic remission, I think that IBD patients treated with biologicals or immunosuppressants should be assessed for endoscopic improvement while on treatment. Did authors perform endoscopy in these patients? Even if in a subset of patients?
Why did the authors choose these tools (Visual Analogue Scale and the Shortened Fatigue Questionnaire) for assessing fatigue? There are other tools available for IBD patients...
In the discussion section of the manuscript, the sentence "A good correlation was observed between the SFQ and fVAS, indicating that the latter can be implemented as a quick screening tool to identify fatigued patients during IBD consultations." does not show any reference. If the sentence is not supported by literature it seems that data reported from this study is not sufficient for this argumentation.
Comparing ADA with IFX for fatigue needs data on the prevalence of fatigue in the two groups before treatment and these data are missing in this work. Are authors sure that they can provide unbiased data on the difference between treatments?
A final remark on language and style: the use of the first person in a scientific paper sound quite inappropriate, I suggest considering using an impersonal form. E.g., "...followed at our tertiary center." in my opinion should be changed to "... followed at a single tertiary referral hospital" or something similar; "...in our study." should be changed to "...in the present study.", etc.
Author Response
Point 1: I don't understand why the authors selected only patients treated with immunosuppressants or biologicals; they do not represent the whole IBD population. These criteria should be changed or explained and the paper title changed accordingly. At first reading, it seems that authors used convenience sampling, like if the authors were able to recruit and follow-up only patients with these characteristics.
Response 1: The goal of the current study was to assess fatigue in a homogenous study population of patients with IBD. Since fatigue is related to disease activity and possibly to disease severity, we wanted to select, as much as possible, a group of patients with comparable disease severity and activity. Therefore, long-term clinical remission under treatment with biologicals and immunosuppressants was used as criteria. As perceptively suggested by the reviewer the title was adjusted accordingly.
Point 2: Did the authors perform CRP or HS-CRP? In my opinion, the latter should be better for IBD patients. Please add an explanation.
Response 2: In this cohort we performed routine CRP measurements and not HS-CRP. Additionally, serum lipocalin-2 levels were assessed and confirmed disease remission.
Point 3: Intestinal inflammation biomarkers should be added to screen patients in order to confirm disease remission (e.g. Faecal Calprotectin or Lactoferrin). In my opinion, symptoms and blood tests are not enough. Talking about endoscopic remission, I think that IBD patients treated with biologicals or immunosuppressants should be assessed for endoscopic improvement while on treatment. Did authors perform endoscopy in these patients? Even if in a subset of patients?
Response 3: We agree with the reviewer that it would have been useful to include fecal calprotectin and endoscopic data to confirm disease remission. However, since we assessed a real-life cohort in a non-interventional study we did not require fecal calprotectin and endoscopic evaluation for every patient (only according to investigator’s decision). In this cohort fecal calprotectin was available from 36 patients, with a median calprotectin of 29.5 mg/kg (IQR 14-54), confirming clear biochemical remission in this small subgroup of patients. Of only 29 patients an endoscopic evaluation was performed in the 6 months before or after inclusion of whom 21 were in clear endoscopic remission (72.4%). In 5 additional patients and endoscopic response was seen (17.2%). Even though these are small subgroups of the total population this further confirms a state of disease remission in the majority of patients.
Point 4: Why did the authors choose these tools (Visual Analogue Scale and the Shortened Fatigue Questionnaire) for assessing fatigue? There are other tools available for IBD patients...
Response 4: For the current study the VAS and SFQ were selected due to the fact that these are both short and easily implementable questionnaires for daily clinical practice. We added the benefits of using these tools to the manuscript.
Point 5: In the discussion section of the manuscript, the sentence "A good correlation was observed between the SFQ and fVAS, indicating that the latter can be implemented as a quick screening tool to identify fatigued patients during IBD consultations." does not show any reference. If the sentence is not supported by literature it seems that data reported from this study is not sufficient for this argumentation.
Response 5: We added other references suggesting a good correlation between fVAS and other longer fatigue questionnaires, thus supporting the evidence that fVAS might be a good screening tool for fatigue.
Point 6: Comparing ADA with IFX for fatigue needs data on the prevalence of fatigue in the two groups before treatment and these data are missing in this work. Are authors sure that they can provide unbiased data on the difference between treatments?
Response 6: As indicated by the reviewer, it would have been interesting to include a fatigue screening before the initiation of biologicals, however, the goal of the current study was not to do a longitudinal assessment of fatigue but to evaluate the prevalence of fatigue despite the induction of a long-term clinical remission. Therefore, we would like to refer to the study of Borren et al. of 2019, assessing the longitudinal trajectory of fatigue, in which no significant difference was found in the percentage of patients with fatigue at the initiation of different biologic treatments.
Point 7: A final remark on language and style: the use of the first person in a scientific paper sound quite inappropriate, I suggest considering using an impersonal form. E.g., "...followed at our tertiary center." in my opinion should be changed to "... followed at a single tertiary referral hospital" or something similar; "...in our study." should be changed to "...in the present study.", etc.
Response 7: As for the language and style, the manuscript was adapted according to the reviewer's suggestions.
Round 2
Reviewer 3 Report
Thank you for giving me the opportunity to read the revised manuscript entitled "Prevalence of fatigue and subclinical depression in patients with inflammatory bowel disease in remission: an observational monocentric study".
As previously stated, the work is interesting and adds information on the issue of fatigue in the IBD population. The comments I made on the manuscript and that were reported have been resolved in this new version that, in my opinion, is now more clear and can be accepted for publication without any other comments or revision required.